# Evolution of a Human-Specific De Novo Open Reading Frame and Its Linked Transcriptional Silencer

**DOI:** 10.3390/ijms25073924

**Published:** 2024-03-31

**Authors:** Nicholas Delihas

**Affiliations:** Department of Microbiology and Immunology, Renaissance School of Medicine, Stony Brook University, Stony Brook, NY 11794, USA; nicholas.delihas@stonybrook.edu; Tel.: +1-631-286-9427

**Keywords:** evolution of transcriptional silencer, exonic silencer, molecular evolution, de novo ORFs, gene evolution, biased mutations

## Abstract

In the human genome, two short open reading frames (ORFs) separated by a transcriptional silencer and a small intervening sequence stem from the gene *SMIM45*. The two ORFs show different translational characteristics, and they also show divergent patterns of evolutionary development. The studies presented here describe the evolution of the components of *SMIM45*. One ORF consists of an ultra-conserved 68 amino acid (aa) sequence, whose origins can be traced beyond the evolutionary age of divergence of the elephant shark, ~462 MYA. The silencer also has ancient origins, but it has a complex and divergent pattern of evolutionary formation, as it overlaps both at the 68 aa ORF and the intervening sequence. The other ORF consists of 107 aa. It develops during primate evolution but is found to originate de novo from an ancestral non-coding genomic region with root origins within the Afrothere clade of placental mammals, whose evolutionary age of divergence is ~99 MYA. The formation of the complete 107 aa ORF during primate evolution is outlined, whereby sequence development is found to occur through biased mutations, with disruptive random mutations that also occur but lead to a dead-end. The 107 aa ORF is of particular significance, as there is evidence to suggest it is a protein that may function in human brain development. Its evolutionary formation presents a view of a human-specific ORF and its linked silencer that were predetermined in non-primate ancestral species. The genomic position of the silencer offers interesting possibilities for the regulation of transcription of the 107 aa ORF. A hypothesis is presented with respect to possible spatiotemporal expression of the 107 aa ORF in embryonic tissues.

## 1. Introduction

*SMIM45* is a complex gene, and its components are only partially understood. The gene encodes an evolutionary ultra-conserved 68 amino acid (aa) ORF (The UniProt Consortium 2023). This ORF is close to a 107 aa ORF, a sequence that is only present in humans [1]. The two ORFs are separated by a small intervening sequence and a transcriptional silencer; the silencer overlaps part of the intervening sequence and also part of the 68 aa ORF. Transcriptional silencers are segments of DNA that suppress the transcription of a target gene. Although their existence has been known for a long time, these genomic regulatory units are just beginning to be studied in detail [2].

Analyses show that a *SMIM45* RNA transcript that carries the three components of a 68 aa ORF, silencer, and 107 aa ORF is transcribed in somatic brain tissues [3]. Additionally, in somatic cells, there is evidence for translation of the 68 aa ORF based on Ribo-Seq ribosome profiling, but not for the 107 aa sequence [3,4] (also see Section 4). Currently, the 107 aa ORF is unannotated by the NCBI due to a lack of evidence of translation in somatic tissues. Other protein identification techniques, mass spectrometry, and polyclonal antibody analyses show a protein product for the 107 aa ORF under certain conditions [5], and it has been proposed that there may be restricted expression of the 107 aa ORF in specific embryonic cells at specific times [6]. In an innovative study, An et al. [6] discovered a protein that functions in human brain development, and it was proposed to be the 107 aa ORF of *SMIM45*. However, continued investigations are needed to show the spatiotemporal expression of the 107 aa ORF in embryonic tissues.

In terms of the evolution of *SMIM45*, the 107 aa ORF is human specific but has de novo origins in an ancient ancestor. De novo protein genes are defined as genes that originate from a non-coding genomic region of an ancestral genome, i.e., “from scratch”. Contrary to the past views that new genes are primarily formed by gene duplication [7], a large body of evidence accumulated during the past 15–20 years shows that a significant number of protein genes originate via de novo means [8,9,10,11,12,13,14,15,16,17,18]. The ancestral genomic regions that contain no coding capacity are assumed to be random sequences. The formation of de novo ORFs from a non-coding region can be visualized as an initial start with a selected root base pair (bp) or bps to be evolutionarily fixed. However, the detection of the evolutionary root bp of an ORF is exceedingly difficult, as distinguishing root DNA base pairs from presumed random sequences is very problematic. Attempts to find root bps in ancestral genomic DNA that have no coding capacity have not been reported. However, the evolutionary progression of an ORF that carries a transmembrane motif from an ancestral non-coding sequence has been described for the yeast genomic locus YBR196C-A [17], and Vakirlis et al. [19] report the evolutionary formation of a class of microproteins in primates where early developmental stages and species of origin have been proposed.

The work presented here traces the evolutionary development of the major components of *SMIM45*—the 68 aa ORF, the transcriptional silencer, and the 107 aa ORF—with a primary focus on the silencer, the 107 aa ORF, and de novo origins. We described the use of genomic DNA sequences and their translated ORFs to analyze evolutionary development. We find that the formation of the 107 aa ORF occurs due to biased random mutations. In addition, a search for the root species of origin of the 107 aa ORF, as well as the segment of the silencer that overlaps with the intervening sequence, shows the Afrothere clade of placental mammals as the root species of origin for both the ORF and the silencer. Thus, the segment of the silencer that overlaps with the intervening sequence and the 107 aa ORF appear to have evolutionarily co-evolved. Biased random mutations, which are found during the development of the 107 aa ORF, refer to the biased selection of a bp from a mutation where the bp is fixed during evolution but with random mutations also occurring. This offers the interesting question of what the nature of the information stored in the Afrothere ancestral genome that fixes base pairs and evolutionarily results in a human-specific ORF is. With respect to the molecular genetics of the *SMIM45* gene, there are significant open questions. We discuss the transcriptional and translational aspects of the two *SMIM45* ORFs, with a proposal involving the repression and activation of transcription of the 107 aa ORF by the silencer.

## 2. Results

### 2.1. Characterization of the Human SMIM45 Gene

In the work presented, it was possible to follow the evolution of DNA that leads to the formation of the human *SMIM45* 68 aa and 107 aa ORFs because of the extreme evolutionary conservation of the 68 aa nt sequence. The 68 aa sequence was used as a guidepost for detecting the 107 aa nt sequence in various species. In addition, the synteny displayed in the chromosomal region containing the ORF sequences also served as a guidepost. *SMIM45* shows synteny with two neighbor genes, *CENPM* and *SEPTIN3*, with *SMIM45* situated in between (Figure 1a,b). The genomic region between these two genes was used to find the *SMIM45* sequence in species where *SMIM45* has not been annotated.

A representation of *SMIM45* is given in Figure 1c. The gene contains 11,991 bp and carries two exons. Exon 1 is flanked by two enhancers, but the exon has not been characterized. Exon 2 comprises two ORFs that are separated by a small intervening 324 bp sequence and a transcriptional silencer. The silencer is termed LOC130067579, ATAC-STARR-seq lymphoblastoid silent region 13815. It overlaps both with a segment of the intervening sequence and with a segment of the 68 aa ORF. Approximately 2.6% of silencers are found to overlap protein coding regions based on an analysis of one percent of the human genome [20]; thus, overlapping sequences may be relatively rare. A translation of the intervening sequence shows that it is devoid of methionine and an open reading frame.

### 2.2. Ultra-Conservation of 68 aa ORF

To determine the origins of *SMIM45* and how its sequence evolved, an analysis was performed on the evolutionary formation of the gene through its two ORFs present in exon 2. The 68 aa ORF displays a remarkable conservation over evolutionary time (Figure 2). There is only one amino acid residue difference between the mouse (*Mus musculus*) and human sequences. The aa sequence from the tarsier [Philippine tarsier (*Carlito syrichta*)], a lower primate and member of the prosimians, shows 100% identity with the human sequence. The Philippine tarsier primate evolutionarily diverged from a common ancestor ~58 million years ago (MYA) and the mouse ~87 MYA. This degree of ultra-conservation during primate evolution may show that any aa change may compromise gene function in primates. The 68 aa sequence has been traced as far back as the elephant shark (*Callorhinchus milii*), with an evolutionary age of divergence of ~462 MYA (*Ensembl* annotation, ENSCMIG00000000567.1). The bottom of Figure 2 shows an alignment of the elephant shark aa sequence with that of the human 68 aa and displays a 65% aa identity. Further detection of the *SMIM45* sequence in evolutionary time was difficult. Both the European River lamprey (*Lampetra fluviatis*) (divergent time ~563 MYA) and the sponges (*Porifera*) (divergent time ~758 MYA) have not been annotated for *SMIM45*, *CENPM*, or *SEPTIN3*. Thus, the mode of formation of the 68 aa sequence via a de novo process or through gene duplication is difficult to assess.

The C-terminal end of the 68 aa sequence that contains the last twelve aa residues shows significant differences in its identity relative to the rest of the 68 aa sequence (Figure 2, bottom), 42% vs. 70%. The C-terminal aa sequence alignment with the human sequence also shows no significant E value (the probability of two sequences aligning by chance). The DNA sequence of the last twelve aa at the C-terminal end also overlaps with the sequence of the silencer (Figure 1c and Appendix A); thus, the difference in identity may relate to differences in the evolution of the overlapping silencer sequence compared to that of the rest of the 68 aa. The presence of the overlapping sequence suggests a dual role for the C-terminal end, one at the aa level concerning the function of the translated 68 aa ORF and the other at the genomic DNA nt level concerning the function of the silencer. Thus, there appear to be two developmental constraints in the evolutionary formation of the C-terminal end: the formation of an aa sequence necessary to form a functional 68 aa ORF and the development of the DNA sequence to form a functional silencer.

Concerning the properties of silencers, there are three sequence motifs that are common to human silencers [20]. These are present in the silencer LOC130067579, but they are outside of the segment of the silencer that overlaps with the 68 aa ORF (see Section 4). LOC130067579 also carries the motif CTCCC, which is the binding sequence for the transcription factor RFLAT-1/KLF13, and this also is outside of the overlapping sequence [21]. As of yet, there is no information on the functions of silencers that overlap ORFs [20]. The silencer that overlaps with the 68 aa ORF may fall into a special category of what may be called exonic silencers, silencer sequences that overlap with ORFs, just as a special category of enhancers, exonic enhancers, are known to overlap with coding sequences [22].

### 2.3. Evolution of the Transcriptional Silencer

The transcriptional silencer also overlaps with the intervening sequence (Figure 1c and Appendix A). An analysis of the evolutionary histories of the two overlapping sections of the silencer shows differences. For simplicity, we will call the transcriptional silencer sequence that overlaps with the 68 aa ORF ‘silencer1’ and the overlap with the intervening sequence ‘silencer2’.

The genomes of four species were analyzed to determine the presence of sequences homologous to silencer1: elephant shark (~462 MYA), reedfish (~429 MYA, frog (~352 MYA), and turtle (~319 MY). The elephant shark and redfish *SMIM45* genes show no significant identity with silencer1, display only random alignments, and show no synteny with the 68 aa ORF sequence (Appendix A). The alignments of the frog and turtle *SMIM45* gene sequences do show positional synteny with the 68 aa ORF sequence, and sequences homologous to silencer1 display relatively high identities: 69% and 71%, respectively (Appendix A). However, there is no significant Expect value with alignments from either species. These are relatively short sequences, and a short sequence has a higher probability of giving a high E value (low significance). Thus, the small sequence length of silencer1 may contribute to the lack of a significant Expect value. In addition, the frog and turtle species are in a gray area or twilight zone [23,24]. As we are looking at species that evolutionarily approach the root species of origin (the reedfish or a related species), there is a decreasing similarity with the human silencer1, and the sequences that show similarities become shorter and shorter. With root species, there would be no sequence identity except for individual root bps. However, with the frog and turtle, positional synteny relative to the 68 aa ORF sequence suggests that the alignments shown are not random. The lack of both synteny and sequence similarity of human silencer1 with the elephant shark and redfish *SMIM45* sequences compared with the relatively high identities with the frog and turtle species appears to point to the root origins of silencer1 from a species the evolved between ~352 MYA and ~429 MYA. This contrasts with the age of origin the 68 aa ORF, which originated from a species that predates the elephant shark (~462 MYA). Thus, the evolutionary development of the 68 aa ORF and the C-terminal end appear to differ, and this likely involves the evolutionary origin of silencer1. It should be noted that the complete sequence of silencer1 is present in the cape elephant shrew (~99 MYA) with a high identity.

Silencer1 is 38 bp in length and silencer2 is 192 bp; thus, most of the silencer is within the intervening sequence. The evolutionary development of silencer2 shows a different species of origin from silencer1. *SMIM45* gene sequences from ancestral species were analyzed to search for the identity to the 192 bp silencer2 sequence. The identification of species that display no significant similarity to the 192 bp, as well as closely related species that have significant but marginal or twilight zone identities [23,24], were looked for and considered species close to the root species origin. Table 1 shows the species evolutionary age of divergence, Expect (E) values, percent identities, and the different segments of the 192 bp silencer2 that display identity with the human silencer LOC130067579. No similarity was detected between the cape elephant shrew and the human silencer (Appendix A). However, the cape golden mole displays a significant E value, as do the lesser hedgehog (tenrec) and aardvark species (Table 1 and Appendix A). These four species are closely related and represent a separate branch of the Afrothere clade of placental mammals [25] (Appendix A). Table 1 indicates that the root origin of silencer2 is a species within this branch of the Afrothere clade that dates back to ~99 MYA. This markedly contrasts with the root origin of silencer1, which was found to be from a species that evolved between 352 MYA and 429 MYA. Thus, regarding the 68 aa ORF, silencer1 and silencer2 display very different evolutionary origins. Table 1 also shows that with decreasing evolutionary age of species divergence, the E value decreases, and the segments that show similarities with silencer2 increase in size. In the rhesus, there is significant similarity with the full silencer2 sequence.

### 2.4. 107 aa ORF, Detection of an Early Developmental Sequence

In sharp contrast to the 68 aa ORF, the 107 aa ORF evolved over ~99 million years of evolutionary time. The evidence presented suggests that a sequence representing an early developmental stage of the 107 aa sequence is in a mouse genomic region that has sequence similarity to the human 107 aa nt sequence. An alignment of a segment of the mouse *SMIM45* gene sequence, a segment of exon 2, and the human mRNA that encodes the 107 aa ORF shows a 57.5% nucleotide sequence identity (Figure 3), with one section of the mouse SMIM45 gene sequence (highlighted in brown) displaying an 80% identity. The alignment locates the mouse genomic region that is homologous to the mRNA of the human 107 aa.

A translation of the nucleotide (nt) sequence of human *SMIM45* exon 2 displays the 107 aa ORF (highlighted in light pink, Figure 4a). However, a translation of the mouse genomic nt sequence homologous to the human 107 aa nt sequence encodes no ORFs except for two methionine start codons (highlighted in pink, Figure 4b). These have no identity with the 107 aa ORF, and based on Blast searches, they do not form parts of proteins in any organisms. A lack of a significant open reading frame indicates that the mouse homologous sequence is non-coding. However, we found the mouse nt sequence (highlighted in brown, Figure 3) that translates to the aa sequence SGLE**_*_**VTVYGGGVQKGKT (**_*_** represents a stop codon). This sequence is shown in Figure 4b, highlighted in turquoise, 5′3′ Frame 3. The homolog in humans is also highlighted in turquoise (Figure 4a). The sequences flanking the turquoise highlighted sequence in the mouse genome do not show similarities with proteins from any species, and we consider these to be random sequences. The mouse aa sequence, SGLE**_*_**VTVYGGGVQKGKT shows 67% similarity and an Expect = 1 × 10^−7^ based on its alignment with the human homologous aa sequence. Thus, the mouse aa sequence, SGLE**_*_**VTVYGGGVQKGKT appears to represent a segment of the human 107 aa open reading frame and to constitute an early stage of development in the 107 aa open reading frame formation. As shown in the text below, the aa sequence is found in descendant species of the mouse. This supports the concept of an early developmental sequence that is present in the mouse genome.

Figure 5 outlines the *mouse SMIM45* gene that is homologous to human exon 2. This depicts the genomic region comparable to the human 107 aa nt sequence, which consists of the region encoding the early developmental aa sequence SGLE**_*_**VTVYGGGVQKGK flanked on both sides by random sequences.

Table 2 shows the evolutionary progression of the early developmental sequence in primates and reveals development occurring with random mutations. The formation of the human sequence SGLELVRVCGGGMQRDKT proceeds rapidly; the lemur, a primitive prosimian primate, formed an almost complete sequence corresponding to the human sequence, showing 83% similarity. There is 100% similarity of the *Tibetan macaque* and the baboon (*Papio anubis*) sequences, both Old World monkeys, with the human sequence. However, the evolutionary pattern shows that sequences from several species deviate in similarity, including the tree shrew, tarsier, and gorilla. The tree shrew sequence has a stop codon and several point mutations that are not present in the mouse sequence. The gorilla and tarsier early developmental sequences result from insertions/deletions that produce frameshift mutations. These are random mutations, and as they are not found in descendant species, they are of doubtful significance.

### 2.5. Evolutionary Development of the 107 aa ORF during Primate Evolution

The formation of the 107 aa ORF during primate evolution was analyzed to provide a picture of how the 107 aa sequence grows during evolution in primates. Specific primate species were chosen for the analysis to demonstrate examples of random mutations occurring.

With the mouse sequence (Figure 6), alignment of the translated 5′3′ Frame 3 aa sequence displays the SGLE**_*_**VTVYGGGVQKGKT sequence (highlighted in turquoise) but no other significant contiguous stretches of aa residues with identity to the human 107 aa. There are individual amino acid residues that show similarities, e.g., the mouse _2_CP and _9_F, but alignment in this region of the human 107 aa sequence may be by chance.

With further evolutionary development of the 107 aa, the Philippine tarsier (Carlito syrichta) sequence shows that it has progressed from that of the mouse and displays a methionine start aa residue with four contiguous aa residues of identity that are close to the start codon (Figure 6, tarsier, 5′3′ Frame 1). There is also the beginning of the development of stretches of aa identity in the middle of the sequence and at the C-terminal end (Figure 6, tarsier 5′3′ Frame 3). The presence of segments of similarity in different regions of the tarsier sequence suggests that the development of the 107 aa sequence progresses through the formation of bps in different genomic sections. Of interest, the human aa sequence _73_GGGMQRDKT_81_ (highlighted in green, Figure 6, tarsier, 5′3′ Frame 3), which is part of the early developmental sequence and present in the mouse genome, is absent in the tarsier homolog. This is due to point mutations and the deletion of the mouse nt sequence GGTGTGCA in the tarsier. However, the sequence is present in the human 107 aa RNA (Figure 6, bottom, “Deletion in tarsier”). This represents a random mutation that is specific to the tarsier that is not found in its progeny.

In analyzing the nt sequences of genomic segments of the mouse and lemur [*Lemur catta* (Ring-tailed lemur)] that are homologous to the human 107naa ORF mRNA sequence, in one segment, there are 15 mutations that are evolutionarily fixed (highlighted in purple) in the lemur/human, compared to 4 random mutations (highlighted in yellow, Figure 6, bottom “Biased mutations lemur”). The lemur predates the tarsier by approximately 7 MY; however, both are primitive primates and members of the prosimians. The rationale for choosing this genomic segment is that it shows a high similarity in nt sequence between species. Analysis in other regions of the lemur sequence where there is more sequence variability is difficult, as some bp identities may occur by chance. The alignment in the figure shows both fixed (biased) and random mutations that add additional support for the concept of biased random mutations occurring during 107 aa ORF evolutionary development.

Proceeding further in primate evolutionary development to Old World monkeys, the rhesus sequence shows a significant increase in similarity at the amino terminal end compared to the tarsier. (Figure 7, 5′3′ Frame 1). However, like the tarsier, there is significant development of sequence identity in other sections; the 5′3′ Frame 2 shows larger stretches of contiguous sequence identity in the middle and carboxyl terminal regions. The elongation of these segments shows biased nearest neighbor aa residue additions.

The orangutan sequence displays a further increase in contiguous aa identity in all regions and does not display frame shift mutations (Figure 7). The aa sequence identity compared to the human aa sequence is 79%. However, a 13 bp insertion in the orangutan mRNA, which is not found in the rhesus, chimpanzee, or human mRNAs, results in the addition of the aa sequence **_30_**SMAPAL**_35_** (Figure 7). This is due to a random insertion in the orangutan, but as with the other random mutations mentioned, it is not present in its descendants.

The chimpanzee sequence shows an almost complete human 107 aa ORF, except for the lack of ten amino acid residues at the human C-terminal (VWGERASMGR**107**), (Figure 8, 5′3′ Frame 3). Of interest, this is due to a four bp tandem repeat AGAG that is present in the human sequence but is absent in the chimpanzee nt sequence. This results in a frame shift (Figure 8 Bottom, “**Alignment of chimpanzee nt sequence with human 107 aa mRNA**”), highlighted in turquoise. However, the chimpanzee 5′3′ Frame 2 aa alignment shows the presence of the ten C-terminal aa sequence in the chimpanzee genome. Except for three point mutations, the absence of the small tandem repeat AGAG distinguishes the chimpanzee 107 aa homolog from the human 107 aa. The tandem repeat present in humans may be due to slipped-strand mispairing [26], which occurs at repeat sequences and constitutes a random process. Thus, the major difference between the chimpanzee and human aa sequences appears to be due to a tandem repeat. Levinson and Gutman [26] first proposed the concept of slipped-strand mispairing as a driving force in the evolution of DNA.

Figure 9 is a graphic representation summarizing the evolutionary formation of the 107 aa ORF described in Figure 6, Figure 7 and Figure 8. The open spaces between the NH_2_ and COOH ends represent random aa sequences, and the black dashed lines represent aa residues with identities in the 107 aa sequence. In the mouse *SMIM45* locus, we show the early developmental sequence SGLE**_*_**VTVYGGGVQKGKT flanked by random sequences. In the tarsier, the 107 aa sequence development shows a methionine start and the formation of an aa sequence identity at the NH_2_ and COOH terminal ends. The growth of the 107 aa sequence continues with nearest neighbor additions at three separate regions: the NH_2_ end, the middle containing the early developmental sequence, and the COOH terminal end in the rhesus and orangutan loci. The chimpanzee shows the joining of all segments to provide a contiguous 107 aa sequence, except for the COOH terminal end that is missing 10 aa residues of the human107 aa sequence due to a frame shift and a lack of a tandem four base pair repeat present in the human 107 aa sequence.

It should be noted that analysis using the RepeatMasker Web Server (Version: open-4.0.9, https://www.repeatmasker.org/cgi-bin/WEBRepeatMasker, accessed on 26 February 2024) to search for transposable elements (TE)s shows that the 107 aa ORF contains no TEs. This is consistent with the observed development of the 107 aa ORF by continuous nearest neighbor additions.

### 2.6. Search for Evolutionary Root Species of the 107 aa ORF

Concerning the *SMIM45* gene in species that evolutionarily predate the mouse, partial forms of the human early developmental sequence, SGLELVRVCGGGMQRDKT are found to be present. Thus, this sequence enabled the analysis of the root origins of the 107 aa ORF in species ancestral to the mouse. A display of synteny is important, with the 68 aa ORF location confirming homologous early developmental sequences in ancient species. We consider the lack of both synteny and of a significant nt and aa similarity to the early developmental sequence in a species to be the likely root species of origin or close to the root origin. The *SMIM45* genes of the Afrothere clade of placental mammals [25] (Appendix A) were analyzed, with a focus on the branch representing the aardvark to the cape elephant shrew.

The *SMIM45* gene from the African savanna elephant, *Loxodonta africana* (divergence time, 99 MYA) was first analyzed to search for the presence of the early developmental sequence. A nt sequence homologous to the human early developmental sequence was found, and it displays 79% nt sequence identity with E = 1 × 10^−9^ (Appendix A). The aa sequence shows 57% identity and E = 8 × 10^−5^, but only with a segment of the early developmental sequence, positions 1–14 out of 18 aa (Appendix A) This shows the presence of the early developmental sequence in the elephant *SMIM45* gene. The elephant is part of the Afrothere clade of mammals.

Analyses of *SMIM45* were performed with species belonging to the separate branch of the Afrothere clade: the aardvark, lesser hedgehog, cape elephant mole, and cape elephant shrew (Appendix A). The results show there are sequence similarities to the early developmental sequence, but alignments are in a twilight zone of identity [23,24], where it is uncertain whether alignments represent true relatedness or whether sequences align by chance. However, the added parameter of synteny, the position of the aligned sequence relative to the 68 aa ORF, adds a constraint that provides confidence that the observed alignments represent significant similarities. The aardvark, lesser hedgehog, and cape elephant mole aligned sequences align in regions where there is synteny. With the aardvark, alignment of the nt sequence homologous to the early developmental nt sequence indicates a 57% similarity but shows no significant E value (Appendix A). The aa alignment with the early developmental sequence displays 100% similarity, E = 2 × 10^−6^ but for only 7 aas out of the 18 aas of the early developmental sequence (Appendix A). An alignment with the entire early developmental aa sequence shows 39% similarity but no significant E value. The data, although in a twilight zone, show support for the presence of the early developmental sequence in the aardvark *SMIM45* gene based on sequences aligning in the region of synteny with the 68 aas. The lesser hedgehog (tenrec) and cape golden mole *SMIM45* homologous sequences display similar identities to that of the aardvark (Appendix A). However, the cape elephant shrew *SMIM45* sequence shows no significant similarity to the early development sequence and no synteny (Appendix A). It may constitute the root species or be a species close to the root. The cape elephant shrew is also found to be the probable root or close to the root species of origin of silencer2. This suggests that the 107 aa ORF and silencer2 originated in the same or closely similar species of the Afrothere clade.

A color-coordinated DNA sequence alignment displays bases that are entirely conserved in the early developmental nt sequence during evolution of the Afrothere species (Appendix A). Thirteen positions are completely conserved, including G_10_AA and G_29_G. Might one or more of these correspond to fixed bps in the root species of origin of the 107 aa ORF?

## 3. Discussion

*SMIM45* is a small gene that contains two ORFs and a transcriptional regulator, and these make up less than 1000 bp of genomic sequence. Within this short genomic space, very diverse evolutionary trajectories are displayed. The evolutionary progression of the two *SMIM45* ORFs reveals a sharp contrast in sequence conservation vs. sequence change. The 68 aa ORF, an ultra-conserved microprotein, exhibits strict regulation of bp conservation during evolution; e.g., the ORF shows 18 bps that differ between the mouse and human sequences. However, there is only one bp substitution that results in a change in an aa residue; the other 17 mutations involve the wobble position. This implies a selective pressure that maintains the ORF sequence. The 68 aa open ORF is 100% conserved over ~75 MY during primate evolution. This perhaps indicates an intolerance for change in any amino acid residue in primate species. The root species origin of the 68 aa ORF stems from an organism that predated the elephant shark, before 462 MYA. The ORF has a predicted alpha transmembrane domain (Appendix A) [27], but a protein product has not been isolated and characterized. However, its ubiquitous presence in a wide range of species suggests an essential functional role. The 107 aa ORF presents a different pattern of evolutionary development, which results in a sequence that is present only in humans and is predicted to have a putative function only in embryonic tissues.

Silencer1, the segment of the silencer that overlaps with the 68 aa ORF, evolved differently from the 68 aa and originated from a species whose evolutionary age of divergence is between 352 MYA and 429 MYA. The complete sequence of silencer1 is present in the Afrothere (99 MYA). The function of silencer1 is not known, but as its DNA sequence is highly conserved in the Afrothere clade and descendent species, it may function as a DNA binding site during evolution of silencer2. The evolutionary origin of silencer2 is in the Afrothere clade, and silencer1 could bind factors involved in bp fixing. Silencers are known to provide binding sites for transcriptional regulatory factors [2]. Silencer1 falls into the category of what we call exonic silencers, or silencer sequences that overlap ORFs, but as yet, no information is available on the functional roles of exonic silencers. Many open questions remain concerning transcriptional silencers and their categories and functions [2,20].

The 107 aa ORF sequence originated in the Afrothere clade with the formation of the early developmental sequence. A model for the evolutionary development of the 107 aa sequence may be visualized as an initial selection of a bp in a genomic region of root species that displays a random sequence. This would be a putative starter seed bp that is evolutionarily fixed. The next stage consists of nearest neighbor sequence growth, wherein the example of the mouse *SMIM45* sequence forms the bp sequence encoding the aa sequence SGLE**_*_**VTVYGGGVQKGKT. In primates, other putative root bps in different regions of the *SMIM45* gene are also fixed, and these expand with nearest neighbor additions. The full human 107 aa sequence forms when all nearest neighbor positions are joined. Thus, growth is not by continuous elongation from one region, e.g., the N-terminal. There is bias in seed base pair selection and in nearest neighbor sequence growth. During the development of the 107 aa ORF, random mutations also occur that result from point mutations and frame shifts. However, these are not found in descendant species. Regulatory elements dictate the initial seed bps, the specific nearest neighbor formations, and the protection against deleterious random mutations being carried to descendants. The question arises as to what is the signaling mechanism that determines the fixing of a particular bp in ancient species that evolutionarily leads to formation of the human specific 107 aa ORF. We assume the information required to fix a bp is in the Afrothere genome. Is the DNA repair process involved in correcting a mutated bp? The fixing of bps or their repair would not appear to have survival value for root or descendent species, as there is no functional product formed yet.

The partial evolutionary progression of several protein genes in primates was previously described [18]. One example is the *MYEOV* (myeloma overexpressed) gene that shows a frameshift and one base pair change in the first codon, creating a Met initiation at the N-terminal end that distinguishes the chimpanzee sequence from that of the human sequence. The data were not presented for the determination of a possible early developmental stage for this or other human-specific genes studied, or for possible root origins.

Important questions remain concerning the transcription and translation of *SMIM45*. What is known is that the *SMIM45* transcript NM_001395940.1, transcript variant 1 mRNA represents a transcriptional readthrough of the 68 aa ORF, silencer, intervening sequence, and the 107 aa ORF. In somatic brain tissues, only the 68 aa ORF is translated and not the 107 aa ORF [3]; thus, there is no translational readthrough. This raises the question of why the silencer, intervening, and 107 aa sequences are part of the mRNA. Perhaps the transcript NM_001395940.1 also functions as a lncRNA with the silencer, intervening, and 107 aa ORF sequences, performing roles at the RNA sequence and secondary, tertiary structural domains? There are analogous examples. Many lncRNAs carry small ORFs that are translated, and these ORFs are evolutionarily conserved [28]. These are properties that are similar to those of the transcript NM_001395940.1, at least with respect to the translation of the 68 aa ORF.

Does the human silencer LOC130067579 regulate 107 aa ORF transcription? Although the properties and functions of silencers are just beginning to be understood [2], over 50% of known silencers are in transcriptionally repressed genomic regions and interact with gene promoters that have no expression [29]. Silencers are known to act on proximal genes and may function in specific cell types [20]; they are also known to act as enhancers in different cell types [2,30,31]. In view of these properties, the *SMIM45* silencer may function as a cis-regulatory element, acting on the 107 aa ORF sequence and suppressing its transcription. We propose the suppression of a separate transcription of the 107 aa ORF by the silencer in somatic cells, independent of the transcript NM_001395940.1. In certain embryonic cells and at certain times during embryonic development, the silencer may function as an enhancer, resulting in the specific transcription and subsequent translation of the 107 aa ORF in embryonic cells. The proposed concept dovetails with the mass spectrometry and polyclonal antibody data showing the translation of the 107 aa protein [5] and the proposal of the spatiotemporal expression of the 107 aa ORF in neuro embryonic tissues [6]. There is a precedent for the spatiotemporal expression of genes in embryonic neuronal cells of different species and over evolutionary time. In looking at the expansion of the neocortex, a study of gene expression in neuronal cells of different species showed that many genes lost or gained expression during embryonic development [32]. The above hypothesis is an attempt to explain unknown aspects of the transcription/translation of the 107 aa ORF sequence. This concept is consistent with known silencer functions and the gain or loss of gene expression that occurs in embryonic development. Experimental data are needed to determine if the hypothesis is correct.

## 4. Materials and Methods

### 4.1. Protocol for Finding the Evolutionary Early Developmental Stage of the 107 aa ORF

The genomic region of the mouse *SMIM45* that contains the 68 aa ORF displays synteny with the genomic region containing the human 107 aa ORF nt sequence. This serves as a guidepost for locating genomic nt sequences that are homologous to the human 107 aa nt sequence in the mouse and other species. By aligning the *SMIM45* gene sequence from the mouse and other species with the sequences of human *SMIM*45 exon 2 and the human 107 aa mRNA, the mouse *SMIM45* region that has homology with the human 107 aa mRNA is determined. The EMBL-EBI Clustal Omega alignment program (https://www.ebi.ac.uk/Tools/msa/clustalo/, accessed on 26 February 2024) is used for nt and aa sequence alignments. When studying a human gene that is not annotated in an ancestral species, the entire genomic sequence between two flanking genes that display synteny can be analyzed for similarities with the gene nt sequence. This is described below for *SMIM45*. Adjacent evolutionary conserved sequences are important to use as guideposts. With other genes, highly conserved enhancer sequences, when found adjacent to or within the gene being analyzed, can also be useful as a guidepost in locating a homologous sequence in an ancestor. For *SMIM45*, *SEPTIN3* served as the consistent guidepost.A segment of the mouse *SMIM45* nt sequence that produces a stretch of similarity to the human 107 aa mRNA is used to translate the nt sequence using the Expasy translation tool. One of the resulting aa sequences from the three translated 5′3′ Frames is found to contain the early developmental sequence. Thus, by aligning the translated three 5′3′ Frame sequences from the mouse or primate species with the human 107 aa ORF aa sequence, the early developmental aa sequence homologous to the human 107 aa sequence is obtained.For primates, as the growing aa sequence evolves during evolution of the ORF, the 5′3′ Frame aa alignments can be used identify additional aas that have an identity with the 107 aa sequence. With the mouse genomic sequence, only random identities are found outside of the early developmental aa sequence.For species that do not have the *SMIM45* gene annotated (e.g., the tarsier), the *SMIM45* sequence was determined by analyzing the DNA sequence between the two genes that show synteny, *SEPTIN3* and *CENPM*. Nt sequences of the genomic regions are then aligned with the 68 aa sequence. This gave an accurate location of the homologous 107 aa nt sequence and/or the early embryonic sequence in these species.To assess possible species that may represent root origins of the 107 aa ORF, entire *SMIM45* gene sequences from various species were aligned with the nt sequences of the 68 aa, 107 aa, and the human early developmental sequence SGLELVRVCGGGMQRDKT. Genomes of ancestral species that show no identity to the early developmental sequence and species that show a marginal but significant similarity are considered candidates for root origins. Synteny with the 68 aa sequence served as a guide in pinpointing sequences that are homologous to the early developmental sequence.

#### Species Analyzed for 68 aa and/or 107 aa ORFs

*Callorhinchus milii*, (elephant shark), *Bombina bombina*, (the fire-bellied toad), *Erpetoichthys calabaricus* (reedfish), *Xenopus tropicalis* (tropical clawed frog), *Caretta caretta* (Loggerhead turtle), *Phascolarctos cinereus* (koala), *Monodelphis domestica* (gray short-tailed opossum), *Elephantulus edwardii* (Cape elephant shrew), *Chrysochloris asiatica* (Cape golden mole), *Orycteropus afer afer* (aardvark), *Echinops telfairi* (small Madagascar hedgehog), *Elephas maximus indicus* (elephant), *Talpa occidentalis* (Iberian mole), *Bos taurus* (cattle), *Leopardus geoffroyi* (Geoffroy’s cat), Oryctolagus cuniculus (rabbit), *Mus musculus* (house mouse), Tupaia chinensis (Chinese tree shrew), Lemur catta (Ring-tailed lemur), Carlito syrichta (Philippine tarsier), Macaca mulatta (Rhesus monkey), Macaca thibetana thibetana (The Tibetan macaque), Papio anubis (olive baboon), Pongo abelii (Sumatran orangutan), Gorilla gorilla gorilla (western lowland gorilla), Pan troglodytes (chimpanzee), and Homo sapiens (human) were analyzed.

For species for which *SMIM45* has not annotated by the NCBI, sequences homologous to the *SMIM45* gene were determined by aligning the entire genomic sequence between the genes of synteny, *CENPM* and *SEPTIN3*, with the human *SMIM45* gene sequence. The homologous 68 aa nt sequence was readily visible.

### 4.2. Source and Properties of Gene and Transcript Sequences

The *SMIM45* gene (NCBI Gene ID: 339674) sequence was obtained from the website: home gene NCBI, https://www.ncbi.nlm.nih.gov/gene/?term=smim45+human, accessed on 26 February 2024 [4] (Sayers et al. 2022). The exon 2 sequence is from *Ensembl* https://useast.ensembl.org/Homo_sapiens/Gene/Sequence?db=core;g=ENSG00000205704;r=22:41952150-41958939;t=ENST00000381348, accessed on 26 February 2024 [33]. The *SMIM45* protein data were obtained from UniProtKB/Swiss-Prot (The UniProt Consortium 2023) and the NLM/NCBI [4].

The *SMIM45* exon 2 nt sequence is as follows:    1 ccaaggccgc cgcgatgccg cacttcctgg actggttcgt gccggtctac ttggtcatct  61 cggtcctcat tctggtgggc ttcggcgcct gcatctacta cttcgagccg ggcctgcagg   121 aggcgcacaa gtggcgcatg cagcgccccc tggtggaccg cgacctccgc aagacgctaa   181 tggtgcgcga caacctggcc ttcggcggcc cggaggtctg agccgacttg caaaggggat   241 aggcgggcgg caccgggcgc cctcccccag cccgccccgc ccgcccagcc cggagacccc   301 caaggcagag ggaggccggc ctgttggccc tccacgctat ccctctgcag cctgggccct   361 cccgacagag gccccaggtg cgctggcagt ggaggtgggg cacttaggtg cctggctggc   421 ccagggcttg ctctccgtgt caagccgact cacccagagc ccaccctccc aagctcaggg   481 gcatcctccg ctgggcccca gtgcctttgc gctgcgcagc actctgccct ccactggact   541 caggcatgtc tatggctgcc tgtcctgagg ctccggagcc ctcatttctt cgtgaagtcc   601 ccagctcccc tgcctccact caatggcacc ggccctgcaa ctttaggcag gtcgaagcca   661 acccaaggaa agaacctaag aacctcgttt ggagggatgt cagcttgggc cagaccagcc   721 gcaccccgcg gggctcaggc ttggaactgg tgagggtgtg tggtgggggt atgcagaggg   781 ataagaccgt ggtagaggag agggttggtg aggagagaga gagagagaga gagagagtct   841 ggggggagcg ggcaagcatg gggagatgag atgtgtatat gtgagagaga gtgtgggggc   901 cccaggcagg gcaggaggtg gtggaaacgg ggtgaactcc gtgggctgtg tgaggactgt   961 ccatagtggg tcccaacccc ctccctctgc tggagtttcc tagcccttcc ccctccccaa 1021 gactgtggca gcaggcagga gcccctgccc tccctccctg tcctgtgcca cacttctggg 1081 gccaaaccca gcccccttga gccaggccct gccagactcc aagcccaccc tagaaccctc 1141 ctcctgtgtg gagactctgt tgccccactt tggacacaga ttggcaacct gcctcacccc 1201 gccccccttc gctggggctt ccatcttaat ttattctcaa taataaagac ttcatgatga 1261 tctctgca

It should be noted that the 5′ end sequence of the annotated human gene *SMIM45*, Ensembl SMIM45 ENSG00000205704, is 4602 bp shorter than the annotated NCBI SMIM45 Gene ID: 339674. However, the exon2 sequence is in both annotations.

The deduced mRNAs that encode the human 68 aa ORF and 107 aa ORF are as follows:68 aa mRNA
     1 atgccgcact tcctggactg gttcgtgccg gtctacttgg tcatctcggt cctcattctg   61 gtgggcttcg gcgcctgcat ctactacttc gagccgggcc tgcaggaggc gcacaagtgg 121 cgcatgcagc gccccctggt ggaccgcgac ctccgcaaga cgctaatggt gcgcgacaac 181 ctggccttcg gcggcccgga ggtctga
107 aa mRNA
     1 atgtctatgg ctgcctgtcc tgaggctccg gagccctcat ttcttcgtga agtccccagc   61 tcccctgcct ccactcaatg gcaccggccc tgcaacttta ggcaggtcga agccaaccca 121 aggaaagaac ctaagaacct cgtttggagg gatgtcagct tgggccagcc cagccgcacc 181 ccgcggggct caggcttgga actggtgagg gtgtgtggtg ggggtatgca gagggataag 241 accgtggtag aggagagggt tggtgaggag agagagagag agagagagag agtctggggg 301 gagcgggcaa gcatggggag atga

The 107 aa mRNA is GC rich: Full Length(324bp) | A(22% 72) | T(19% 57) | G(36% 119) | C(23% 76)

The intervening sequence between the human 68 aa and 107 aa nt sequences (which includes the overlapping sequence) is as follows:     1 gccgacttgc aaaggggata ggcgggcggc accgggcgcc ctcccccagc ccgccccgcc   61 cgcccagccc ggagaccccc aaggcagagg gaggccggcc tgttggccct ccacgctatc 121 cctctgcagc ctgggccctc ccgacagagg ccccaggtgc gctggcagtg gaggtggggc 181 acttaggtgc ctggctggcc cagggcttgc tctccgtgtc aagccgactc acccagagcc 241 caccctccca agctcagggg catcctccgc tgggccccag tgcctttgcg ctgcgcagca 301 ctctgccctc cactggactc aggc

The LOC130067579, ATAC-STARR-seq lymphoblastoid.silent sequence silent region 13815 present in *SMIM45* is as follows:     1 tgcgcgacaa cctggccttc ggcggcccgg aggtctgagc cgacttgcaa aggggatagg   61 cgggcggcac cgggc***gcc***ct cccccagccc gccccgcccg cccagcccgg agacccccaa 121 ggcagaggga ggccggcctg ttggccctcc acgctatccc tctgcag***cct gg***gccctccc 181 gacagaggcc ccaggtgcgc tgg***cagtgg***a ggtggggcac ttaggtgcct

Highlighted in bold and italicized in the above sequence of LOC130067579 are the silencer motifs that are present in the silencers found in K562 cells [20]. The term ‘exonic silencers’, which refers to as an overlap with an ORF, is used here as a parallel to the commonly used term ‘exonic enhancers’ [22].

The 68 aa mRNA nt coordinates are positions 15-221 of *SMIM45* exon 2, and the 107 aa mRNA are positions 546–869. The 107 aa mRNA is a putative mRNA. The nomenclature used for the mRNAs is that shown by the NLM/NCBI.

The NML/NCBI computational analysis of *SMIM45* shows a transcript, NM_001395940.1, Homo sapiens small integral membrane protein 45 (SMIM45) transcript variant 1 mRNA. This transcript is identical to the *Ensembl* transcript ENST00000381348.5 SMIM45-201. The transcript contains both the 68 aa and 107 aa mRNA sequences. UniProt provides evidence for the transcription of ENST00000381348.5 (A0A590UK83 · SMI45_HUMAN). The NLM/NCBI and Ensembl list the 68 aa as a protein product based on the conservation of the ORF and Ribo-Seq elongating ribosome analysis, but there is no protein product for the 107 aa ORF (https://www.ncbi.nlm.nih.gov/genome/gdv/browser/genome/?cfg=NCID_1_31487677_130.14.22.10_9146_1696530571_2728389868, accessed on 26 February 2024). It appears that there is no translational readthrough of transcript NM_001395940.1 that would include the 107 aa ORF. The NCBI has removed the annotation of the 107 aa ORF based on Ribo-Seq and phyloCSF (comparative genomics that are used analyze multispecies of nucleotide sequence alignments); however, the 107 aa ORF is human-specific. Thus, a phyloCSF analysis does not pertain here.

The nt sequence of transcript variant 1, mRNA NM_001395940.1 is as follows:    1 ctctgatggg cagggagaga taccagggtg ctgagccagt ccaggactgc cccctcctgg  61 cccactcaga gcccctgggt gtgagaagct cgtctcccgt gggttgcatt ggctctgccc   121 tatctctgcc tccagcaccc agggcggccg cagatggcag tgtctctggg gacagcagct   181 gcgaatgagt ccacgggcca atgctgagct gctcaggctg aggcggtgtg ctcagcacag   241 agcccccgga actggcatct gcagggcgtg agccaaggcc gccgcgatgc cgcacttcct   301 ggactggttc gtgccggtct acttggtcat ctcggtcctc attctggtgg gcttcggcgc   361 ctgcatctac tacttcgagc cgggcctgca ggaggcgcac aagtggcgca tgcagcgccc   421 cctggtggac cgcgacctcc gcaagacgct aatggtgcgc gacaacctgg ccttcggcgg   481 cccggaggtc tgagccgact tgcaaagggg ataggcgggc ggcaccgggc gccctccccc   541 agcccgcccc gcccgcccag cccggagacc cccaaggcag agggaggccg gcctgttggc   601 cctccacgct atccctctgc agcctgggcc ctcccgacag aggccccagg tgcgctggca   661 gtggaggtgg ggcacttagg tgcctggctg gcccagggct tgctctccgt gtcaagccga   721 ctcacccaga gcccaccctc ccaagctcag gggcatcctc cgctgggccc cagtgccttt   781 gcgctgcgca gcactctgcc ctccactgga ctcaggcatg tctatggctg cctgtcctga   841 ggctccggag ccctcatttc ttcgtgaagt ccccagctcc cctgcctcca ctcaatggca   901 ccggccctgc aactttaggc aggtcgaagc caacccaagg aaagaaccta agaacctcgt   961 ttggagggat gtcagcttgg gccagaccag ccgcaccccg cggggctcag gcttggaact 1021 ggtgagggtg tgtggtgggg gtatgcagag ggataagacc gtggtagagg agagggttgg 1081 tgaggagaga gagagagaga gagagagagt ctggggggag cgggcaagca tggggagatg 1141 agatgtgtat atgtgagaga gagtgtgggg gccccaggca gggcaggagg tggtggaaac 1201 ggggtgaact ccgtgggctg tgtgaggact gtccatagtg ggtcccaacc ccctccctct 1261 gctggagttt cctagccctt ccccctcccc aagactgtgg cagcaggcag gagcccctgc 1321 cctccctccc tgtcctgtgc cacacttctg gggccaaacc cagccccctt gagccaggcc 1381 ctgccagact ccaagcccac cctagaaccc tcctcctgtg tggagactct gttgccccac 1441 tttggacaca gattggcaac ctgcctcacc ccgcccccct tcgctggggc ttccatctta 1501 atttattctc aataataaag acttcatgat gatctctgca

The theoretical RNA sequence that translates to the human conserved early developmental sequence (SGLELVRVCGGGMQRDKT) is as follows:tcaggcttggaactggtgagggtgtgtggtgggggtatgcagagggataagacc

The sequence is G rich: A(20%) T(23%) G(46%) C(11%).

### 4.3. Expect Value (E)

The Expect value (E) relates to the probability of two sequences aligning by chance. The E value decreases with a lower probability of a random alignment.

Together with the display of synteny with the 68 aa ORF, the E value was used to assess the significance of identity between sequences that are homologous to the silencer. For the Afrotherian and other pre-primate species, the following criteria were used to determine significant similarities to the human early developmental sequence: the presence of synteny with the 68aa ORF nt sequence, the expect value, and the percent identity in sequence alignments. As one looks at species close to the root species of origin, the sequence identity and the homologous sequence length become smaller, and analyses are within the “twilight zone” [23,24,34]. The twilight zone has been defined for marginal amino acid similarities of proteins. In the work described here, the twilight similarities for nt sequence alignments were those sequences that showed synteny with the 68aa ORF nt sequence but displayed low sequence identity and high E values; identity of ~55%, E = 0.05 or greater.

### 4.4. Source of Evolutionary Divergent Age of Species

The evolutionary age of divergence of primates and pre-primate species were obtained from references [35,36,37]. The evolutionary age of *Bombina bombina*, (the fire-bellied toad) was obtained from reference [38], the cow from reference [39], and the koala from reference [40]. The Afrothere clade mammals, *Callorhinchus milii* (elephant shark) and other species were obtained from Time Tree of Life, (https://timetree.org/, accessed on 26 February 2024) [41].

### 4.5. Nucleotide and Amino acid Sequence Alignment and Translation Tools

The BLAST program [42] (https://blast.ncbi.nlm.nih.gov/Blast.cgi?PROGRAM=blastn&PAGE_TYPE=BlastSearch&LINK_LOC=blasthome/, accessed on 26 February 2024) was employed using the standard parameters to detect sequence similarity in different species. The Blast two-sequences program (https://blast.ncbi.nlm.nih.gov/Blast.cgi?PAGE=MegaBlast&PROGRAM=blastn&BLAST_PROGRAMS=megaBlast&PAGE_TYPE=BlastSearch&BLAST_SPEC=blast2seq&DATABASE=n/a&QUERY=&SUBJECTS=/, accessed on 26 February 2024) was used to obtain the data presented in Table 1. The program selection was as follows: Optimize for somewhat similar sequence (blastn). The general parameters were as follows: Expect threshold, 0.05. The scoring parameters were as follows: Match/Mismatch, 2,-3; Gap Costs, Extension: 5; Extension: 2; Filters; Low complexity region.

Clustal Omega was employed to align three or more nt or aa sequences [43]. The standard parameters shown on the web page were used. Both EMBOSS Needle and Clustal Omega (Pairwise Sequence Alignment Tools < EMBL-EBI) were used for the alignment of two sequences. In regions where the two sequences were similar, no significant differences were found between the two alignment programs. The data presented in Figure 4, Figure 6, Figure 7 and Figure 8 show alignments identified using Clustal Omega. This was chosen over the alignments identified using EMBOSS Needle, as the Clustal Omega format readily allows for visual identification of aa positions of identity.

The MAFFT online service, multiple sequence alignment, interactive sequence choice and visualization [44] was used to align early developmental sequences and visualize DNA base conservation using color coordination. The ExPASy Translational tool [45] was used for the translation of nucleotide sequences. The Sequence Manipulation Suite [46] was used to obtain reverse complement sequences and for sequence clean up. The RepeatMasker Web Server (https://www.repeatmasker.org/cgi-bin/WEBRepeatMasker, accessed on 26 February 2024) was employed to search for transposable elements in the silencer and the 107 aa ORF sequences.

## 5. Conclusions

Evolutionary analyses of the human *SMIM45* 107 aa ORF indicate that its formation was predetermined ~100 MYA, and its root species of origin is estimated to be from the Afrotherian clade of mammals. Major development of the ORF sequence occurred in primates and proceeded via biased random mutations. Biased mutations imply a directed process of ORF sequence development that may occur via cellular regulatory mechanisms that fix base pairs. However, the origin of the human 107 aa ORF from the Afrothere clade poses the following question: what is the nature of the ancestral DNA that may contain the information required to form the 107 aa ORF, which is in place in these ancient species and with evolutionary time leads to completion of the human ORF? With respect to the transcriptional silencer, the finding of the silencer linked to the 107 aa ORF nt sequence offers interesting possibilities for the regulation of transcription of the107 aa ORF. A hypothesis has been presented that dovetails with the concept of spatiotemporal expression of the 107 aa ORF [6]. Silencer1, the silencer segment that overlaps with the 68 aa ORF sequence, which can be termed an exonic silencer, is present as a complete sequence in the Afrothere, and its sequence remains highly conserved with evolutionary time. As a conserved genomic sequence, it may have a role in sequence development of silencer2, whose origins begin in the Afrothere. The properties of silencers are just beginning to be understood, and it remains to be determined whether silencer1 shares common properties with other exonic silencers.

## Figures and Tables

**Figure 1 ijms-25-03924-f001:**
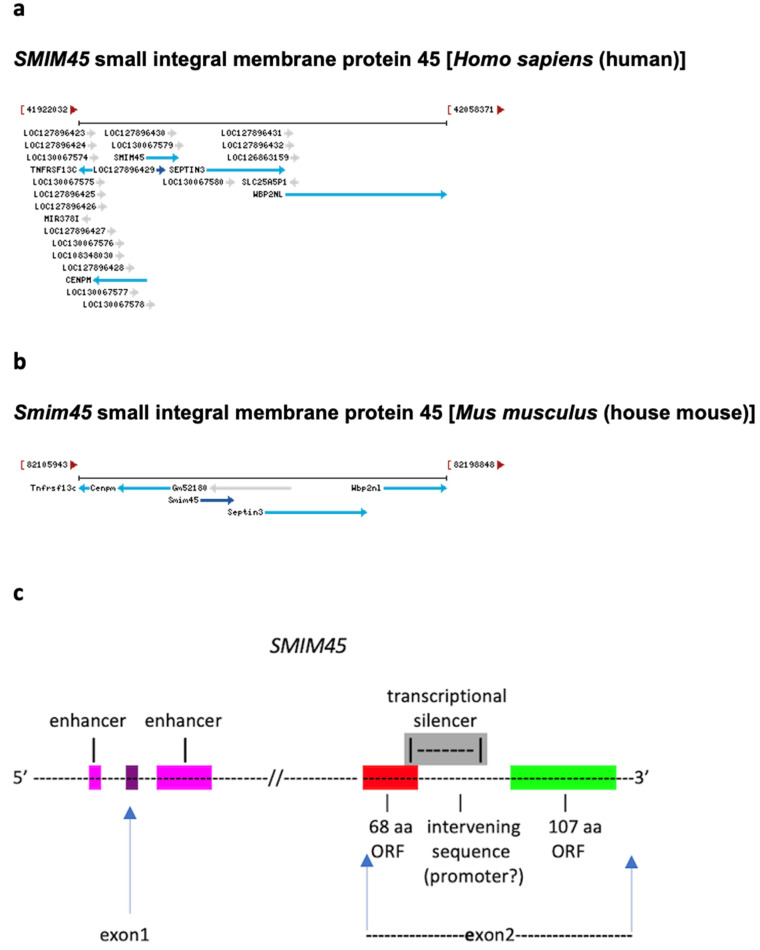
(**a**,**b**) Chromosomal location of *SMIM45* in the human and mouse genomes. The synteny with genes *CENPM* (centromere protein M) and *SEPTIN3* (neuronal-specific septin-3 gene) is shown. Diagrams are from the NLM/NCBI website that describes genes from different species (Home-Gene-NCBI, https://www.ncbi.nlm.nih.gov/gene, accessed on 2 March 2024). The human *SMIM45* gene in the 22Q11.2 chromosomal region displays, in addition to the *CENPM* and *SEPTIN3* genes, a concentration of transcriptional enhancers, transcriptional silencers, microRNAs, ncRNA genes, and other genes. (**c**) A schematic of the human *SMIM45* gene. Shown is the arrangement of the 68 aa ORF, the intervening sequence, the 107 aa ORF, and the overlap of the transcriptional silencer with both the 68 aa and the intervening sequence. The length of the *SMIM45* gene, 11,991 bp, is from the NCBI annotation. The length of the intervening sequence is 324 bp; the silencer sequence, 230 bp; the overlap of silencer with the 68 aa ORF sequence, 38 bp; the overlap of silencer with the intervening sequence, 192 bp; the intervening sequence between the silencer and the 107 aa ORF sequence, 132 bp. The 68 aa ORF carries a predicted transmembrane domain (Appendix A); its nt sequence is GC rich, 63% (Appendix A).

**Figure 2 ijms-25-03924-f002:**
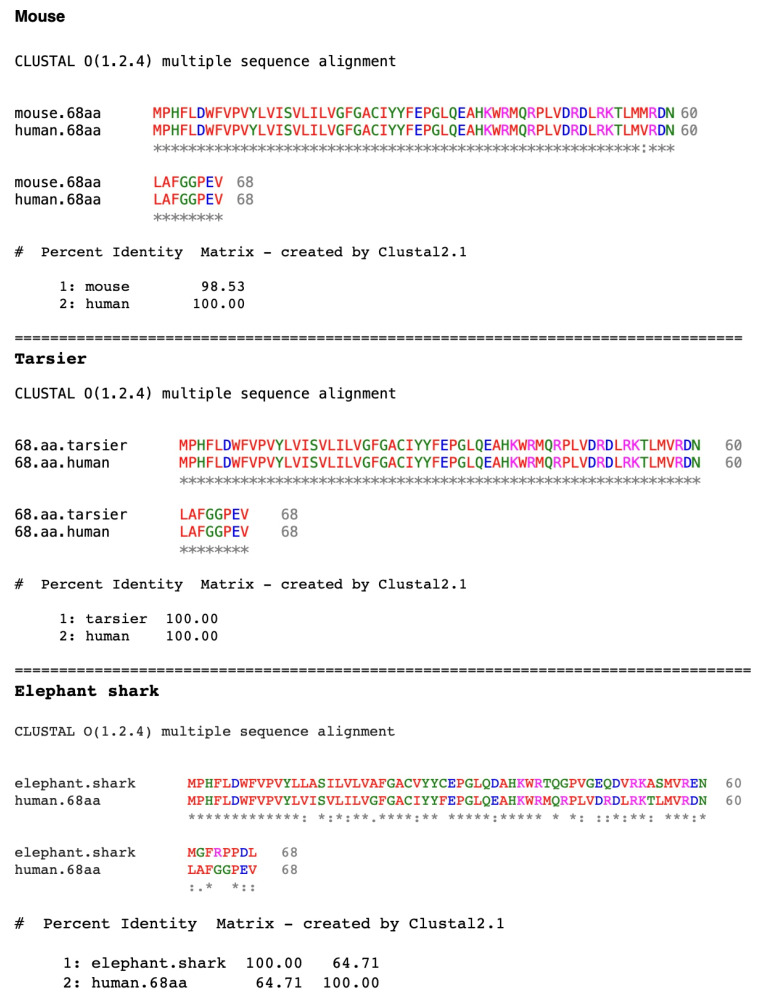
Alignments of the mouse 68 aa ORF with the human 68 aa ORF (**top**), Philippine tarsier (*Carlito syrichta*) (**middle**), and elephant shark (*Callorhinchus milii*) (**bottom**). The translated sequences of the human and mouse *SMIM45* genes are identical except for one aa residue difference (position 57).

**Figure 3 ijms-25-03924-f003:**
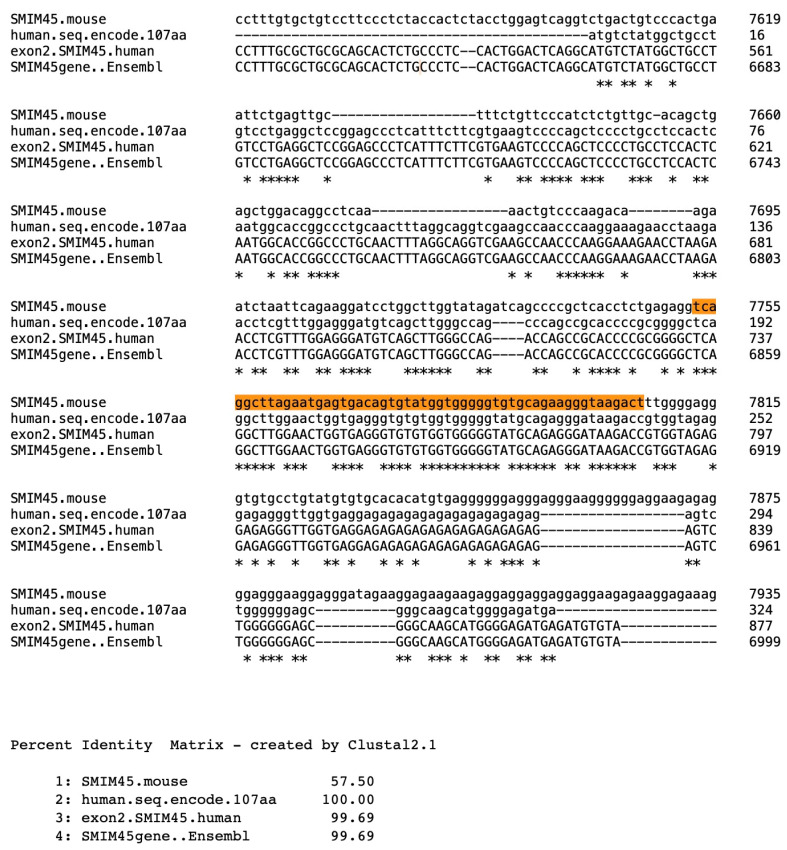
Alignment of a segment of the mouse *Smim45* gene sequence with the human sequence that codes for the 107 aa open reading frame, together with a segment of the nt sequence of exon 2 of the NLM/NCBI annotated human *SMIM45* gene and a segment of the human *SMIM45* gene sequence (as provided by *Ensembl*). The mouse *SMIM45* gene sequence is 8374 bp, as annotated by the NCBI. The alignment was performed for the entire mouse *SMIM45* gene, but only a segment of the sequence is shown in Figure 3.

**Figure 4 ijms-25-03924-f004:**
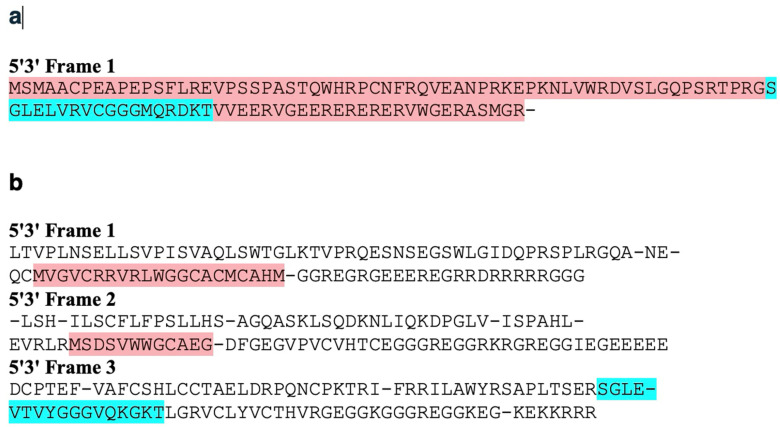
(**a**) Translation of the nt sequence from human *SMIM45* exon 2 107 aa ORF region. The 5′3′ Frame 1 is shown. The light pink highlighted sequence is the 107 aa open reading frame. (**b**) The translation of the mouse genomic region predicted to be homologous to the human 107 aa nucleotide sequence contains the sequence SGLE-VTVYGGGVQKGKT (highlighted in turquoise). The highlighted sequences in light pink are small open reading frames (in 5′3′ Frames 1 and 3).

**Figure 5 ijms-25-03924-f005:**
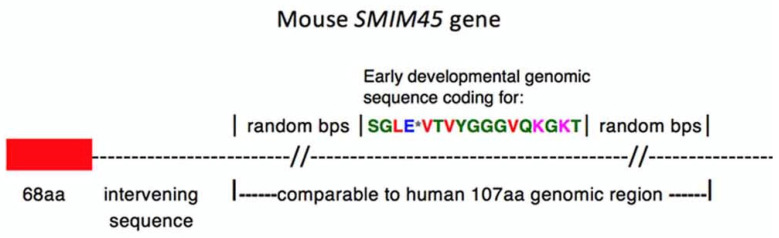
Graphic representation depicting the *mouse SMIM45* gene.

**Figure 6 ijms-25-03924-f006:**
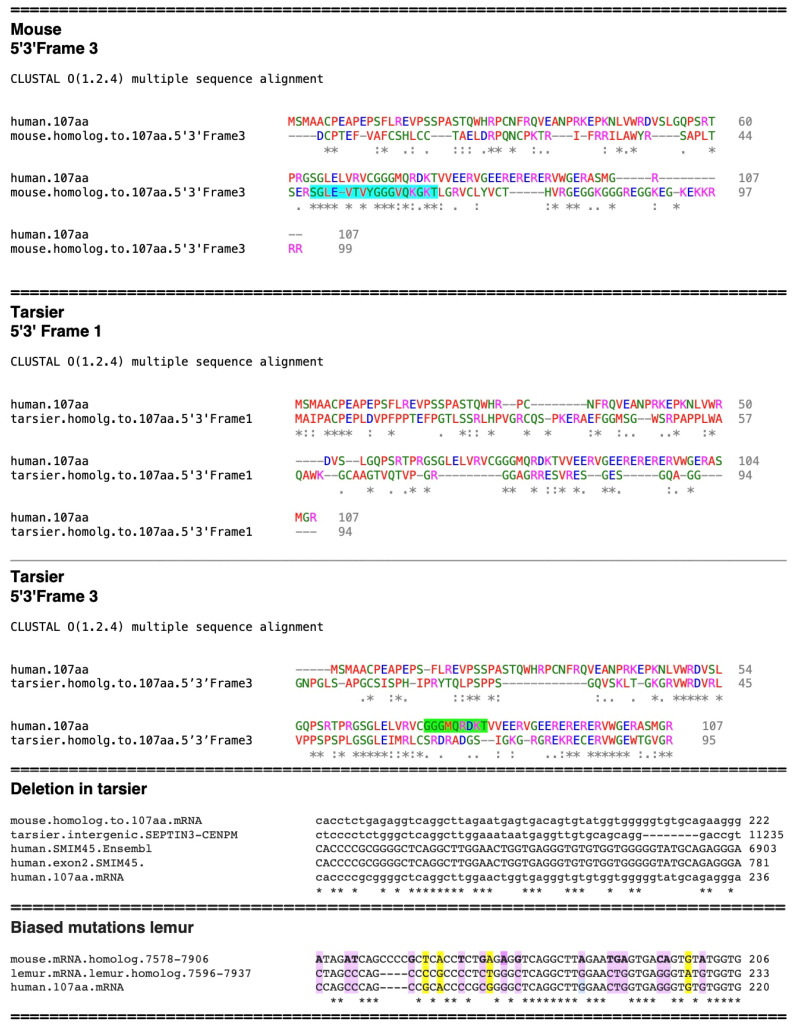
Development of the *SMIM45* 107 aa ORF in the mouse and tarsier. The source of the mouse aa sequences is from reading frames of translated genomic nt sequences homologous to the 107 aa mRNA sequence. The dash in the sequence _48_SGLE-VTVYGGGVQKGKT_64_ refers to a stop codon. The alignment of the human 107 aa sequence with the mouse 5′3′ Frame 3 aa (obtained from a translation of the mouse nt sequence homologous to the 107 aa mRNA) and the alignment of 5′3′ Frames 1 and 3 of the translated tarsier sequence. **Figure 6 bottom**, **“Deletion in tarsier”**. Nt sequence alignment show the deletion of nine bps in the tarsier nt 107 aa sequence. **Figure 6 Bottom**, **“Biased mutations”** shows biased and random mutations between the mouse, lemur, and human sequences.

**Figure 7 ijms-25-03924-f007:**
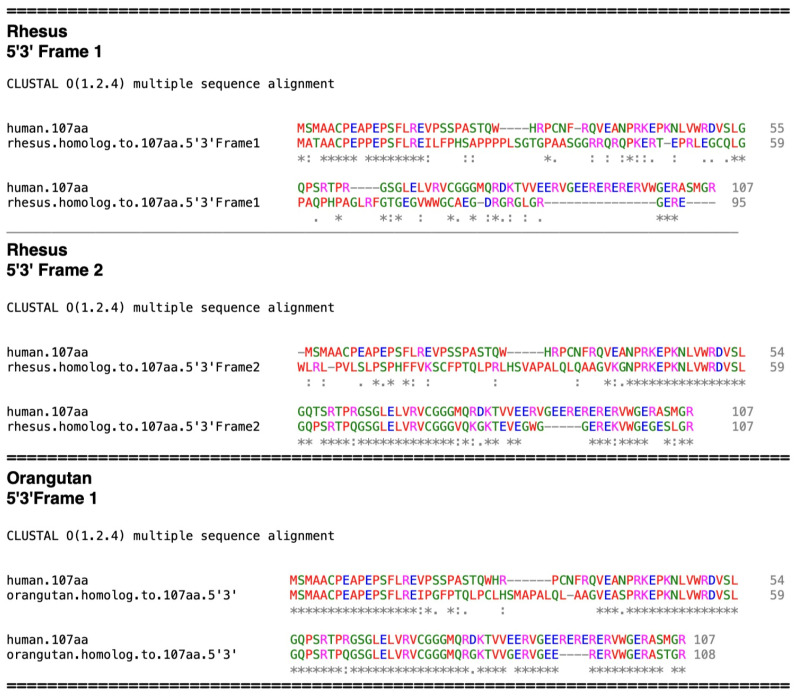
(**Top**) Amino acid alignment of the rhesus sequence homologous to the 107 aa. The 5′3′ Frame 1 and 2 sequences are shown. (**Bottom**) alignment of the homologous 107 aa orangutan sequence with that of the human 107 aa.

**Figure 8 ijms-25-03924-f008:**
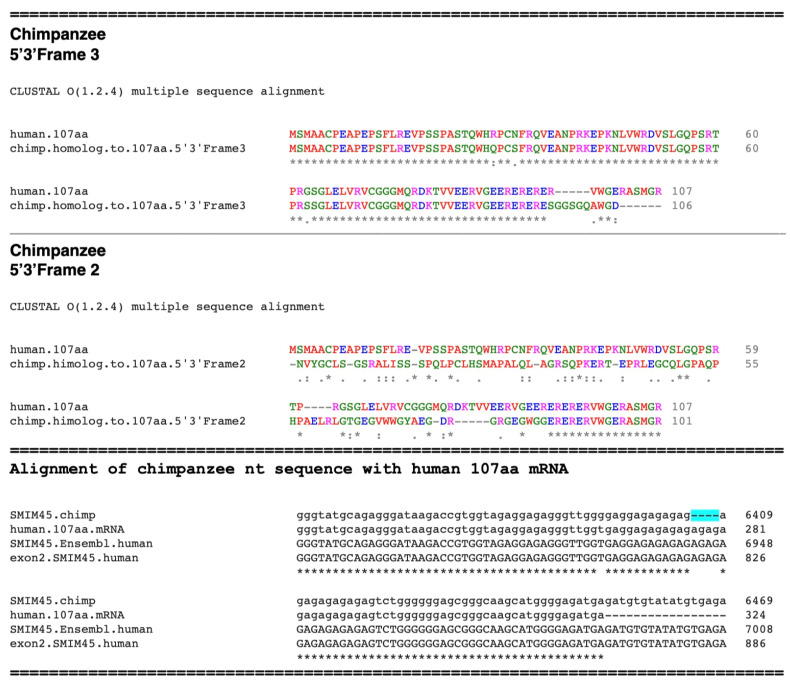
(**Top**) Alignment of the evolving 107 aa sequence in the chimpanzee with that of the human 107 aa. Displayed are **the** 5′3′ Frames 2 and 3 of the chimpanzee homologous sequence. (**Bottom**) An alignment of segments of the chimpanzee *SMIM45* sequence, human *SMIM45* exon 2, the human *SMIM45* sequence from *Ensembl*, and the human 107 aa mRNA sequence. The alignment shows the absence of the 4 bp AGAG tandem repeat in the chimpanzee sequence (highlighted in turquoise). Asterisk, symbols, dot and colon shown are from https://abacus.bates.edu/bioinformatics1/clustalw2.html (accessed on 26 February 2024).

**Figure 9 ijms-25-03924-f009:**
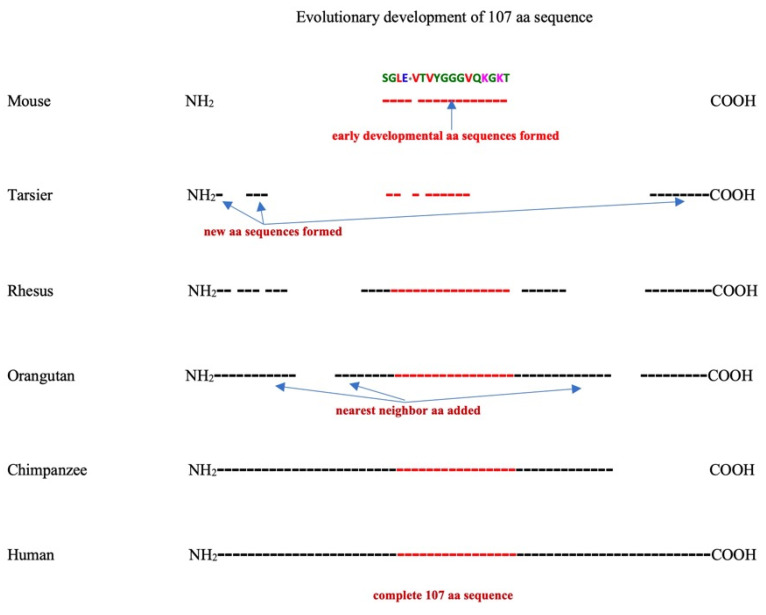
Diagrammatic representation of the evolutionary formation of the human 107 aa sequence. The open spaces represent regions with no aa sequence identity to the 107 aa human 107 aa and are considered random sequences. The dashed lines represent aa residues that have similarities to the human 107 aa sequence. The representation depicts random mutations in the tarsier homolog to the mouse SGLE**_*_**VTVYGGGVQKGKT sequence.

**Table 1 ijms-25-03924-t001:** Evolutionary progression of silencer2 that resides within the intervening sequence.

Species	MYA Age of Divergence from Common Ancestor	Expect	Percent Identity	bp Positions Showing Identity with Silencer2
Cape elephant shrew	99		no significant similarity	
Cape golden mole	99	3 × 10^−13^	78	59–131
Lesser Hedgehog	99	6 × 10^−5^	73	56–100
Aardvark	99	4 × 10^−8^	75	64–131
Elephant	99	8 × 10^−5^	67	35–131
Shrew	94	5 × 10^−7^	86	108–136
Cat	94	1 × 10^−15^	73	28–151
Squirrel	87	1 × 10^−21^	74	54–182
Tree shrew	85	2 × 10^−13^	77	71–152
Lemur catta	61	2 × 10^−25^	71	21–190
Marmoset	43	2 × 10^−40^	84	1–165
Rhesus	31	2 × 10^−70^	88	1–192
Orangutan	20	2 × 10^−75^	92	1–192
Chimpanzee	6	3 × 10^−103^	100	1–192
Human	6	3 × 10^−103^	100	1–192

**Table 2 ijms-25-03924-t002:** Evolutionary progression of the early developmental sequence in primates.

Early Developmental Sequence	Species	% Identity	Evolutionary Age
SG L E _ * _ V T V YGGG V Q K G K T	Mouse	70.5	87 MYA
_ * _ G L E LL R A Y V GG V Q R GCT	Tree shrew	58.82	68 MYA
SG L E LV R V CGGG A Q R G E T	Lemur	83.33	61 MYA
SG L E IM R L CS R D R A D GS	Tarsier	35.29	58 MYA
SG L E LV R V CGGG V Q K G K T	Rhesus	83.33	31 MYA
SG L E LV R V CGGG M Q R D K T	Tibetan macaque	100.00	31 MYA
SG L E LV R V CGGG M Q R D K T	Baboon	100.00	31 MYA
SG L E LV R V CGGG M Q R G K T	Orangutan	100.00	20 MYA
SG L E LV R VWW GY A E G_*_ D R	Gorilla	52.9	10 MYA
SG L E LV R V CGGG M Q R D K T	Chimpanzee	100.00	6 MYA
SG L E LV R V CGGG M Q R D K T	Human	100.00	6 MYA

## Data Availability

Data is contained within the article and Appendix A.

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
