# Peer review of "Evolution of a Human-Specific De Novo Open Reading Frame and Its Linked Transcriptional Silencer"

_ijms, 2024, doi:10.3390/ijms25073924_

Round 1

Reviewer 1 Report

Comments and Suggestions for Authors

In this work, Dr. Nicholas Delihas describes the de novo evolution of an open reading frame found in the human genome that may be expressed during neural development. The evolution of de novo ORFs is of broad interest, especially as it relates to human-specific gain-of-function traits. The author presents well-structured arguments for understanding the relationship between evolutionary pressures at the DNA, RNA and protein level of this short ORF and its flanking silencer sequence, both of which are within a transcribed exon. 

With minor edits to presentation accuracy and additional context of the author's conclusions and interpretation, I would support publication of this manuscript in IJMS. 

Specific comments below:

In the abstract the author should mention that this work focuses on the human genome. In the introduction, the author should briefly summarize previous work that initially identified SMIM45 and the annotation of the two exons. 

This reviewer is confused about the null model presented in section 2.4.107, aa ORF, detection of an early developmental sequence. The author states that 'The probability of a random 54 nt sequence that has 81% identity by chance with the human counterpart is exceedingly small... Thus, the mouse aa sequence appears to represent a segment of the human 107 aa open reading frame and constitutes an early stage of developmental in 107 aa open reading frame formation.' The null model used here for the binomial calculation assumes a completely random sequence choice, which is clearly flawed, as we know there is a nonrandom evolutionary relationship between mouse and human organisms. The more relevant question is whether this sequence has been positively selected in these species at the DNA, RNA or protein level. Can the author speak to this point?

The author mentions that the identified silencer contains the three required pieces of a silencer but doesn't list them or provide evidence. It would help the reader's comprehension to know these three pieces. 

This SMIM45 locus with silencer, exon and 107aa ORF is particularly interesting and allows the author to also comment on the difference in evolutionary pressures for potential 'exonic' silencers that are present in an exon that is transcribed but not translated, versus an exonic silencer that is present in an exon and also within an ORF. 

Relatedly, while the field is particularly excited about the evolution of novel ORFs that have brain-specific function, the evidence that this 107 aa ORF functions in the brain is relatively weak. The author mentions in the discussion that the silencer may shift function to become an enhancer in certain cell types --could the author comment more extensively on the content of the silencer and whether it contains, for example, binding sites for neural-specific transcription factors? That would help bolster this line of thinking.

This reviewer is uncomfortable with the use of diffusive motion of chemotactic bacteria as an analogy for evolutionary direction, especially with the author’s wording that evolution has ‘goals’. These ideas are similar to established models of fitness landscapes, with some mutations moving organisms toward higher or lower regions on a fitness landscape, and the path of moving toward higher fitness not always taking a straight line. This reviewer doesn’t see the need to put forth a new model of diffusive evolution, especially one that invokes ‘goals’ of evolutionary drive, and prefers this section be rewritten.

Regarding the figures, it seems like many of them are screenshots from word processing programs and it is distracting that they have spellcheck error red underlines throughout. It would be beneficial to remove these before publication.

Comments on the Quality of English Language

The written language is clear, but there are some typos scattered throughout. Additionally, the font sometimes shifts back and forth, (e.g. line 31, line 112, line 237) which is distracting for the reader.

A few minor typos are listed below:

Line 24, should be spaces between 'the107aa' -> 'the 107 aa'
Line 117, the period has a strikethrough
Line 283, an extra period
etc.

Capitalization of species is inconsistent, sometimes Orangutan, Chimpanzee, Gorilla are capitalized and other times not.

Reviewer 2 Report

Comments and Suggestions for Authors

A few critical comments can be raised based on the detailed content of the manuscript "Evolution of a human de novo open reading frame and its linked transcriptional silencer" by Nicholas Delihas. These comments are intended to enhance the quality and impact of the study:

1. Clarity and Depth of Evolutionary Analysis

The manuscript presents a comprehensive evolutionary analysis of the SMIM45 gene, mainly focusing on two ORFs (open reading frames) and a transcriptional silencer. The differentiation between the 68 aa ORF's ultra-conservation and the de novo evolution of the 107 aa ORF is intriguing. However, the connection between these evolutionary patterns and their biological implications could be further elucidated. Specifically, how these evolutionary trajectories contribute to the functional divergence or conservation of these ORFs in various species could be more clearly articulated.

2. Experimental Validation of Theoretical Models

The manuscript discusses the theoretical models for the evolution of the 107 aa ORF through biased random mutations alongside an analogy to a biased random walk in chemotaxis. While the analogy provides an exciting perspective, experimental validation of these theoretical models would significantly strengthen the study. For example, experimental data demonstrating the selective pressure or the mechanistic basis for the biased selection of base pairs during ORF evolution could provide concrete evidence supporting the theoretical framework.

3. Functional Characterization of ORFs and Silencer

The study delineates the evolutionary history and formation of the 107 aa ORF and its linked silencer but stops short of a detailed functional characterization. Given the suggestion that the 107 aa ORF may play a role in human brain development, experimental studies that directly investigate the functional implications of this ORF and its regulation by the silencer would be precious. Such investigations could include gene expression analyses in different developmental stages or conditions and functional assays to determine the role of the 107 aa protein in cellular or developmental processes.

4. Methodological Details and Statistical Analysis

The manuscript outlines various methodologies used for the evolutionary analysis, including sequence alignments and the identification of early developmental sequences. However, more detailed methodological descriptions and statistical analyses could enhance the reproducibility and rigor of the study. For example, specifying the criteria for sequence alignment, the thresholds for significance in synteny and sequence identity analyses, and the statistical methods used to assess the probability of sequence identity by chance would be beneficial.

5. Broader Evolutionary Implications

While the manuscript focuses on the SMIM45 gene and its components, discussing the broader evolutionary implications of de novo ORF formation and transcriptional silencers in genome evolution could provide a wider context. This could include comparisons with other genes that have undergone similar evolutionary processes, discussing the potential for de novo genes to contribute to species-specific traits or adaptations, and considering the evolutionary dynamics of transcriptional regulation.

Comments on the Quality of English Language

Minor editing of English language required

Round 2

Reviewer 2 Report

Comments and Suggestions for Authors

The authors have addressed all the conerns